# Phylogeography of Rotavirus G8P[8] Detected in Argentina: Evidence of Transpacific Dissemination

**DOI:** 10.3390/v14102223

**Published:** 2022-10-09

**Authors:** Juan Ignacio Degiuseppe, Carolina Torres, Viviana Andrea Mbayed, Juan Andrés Stupka

**Affiliations:** 1Laboratorio de Gastroenteritis Virales, INEI-ANLIS “Dr. Carlos G. Malbrán”, Avenida Vélez Sarsfield 563, Ciudad de Buenos Aires 1281, Argentina; 2Consejo Nacional de Investigaciones Científicas y Técnicas (CONICET), Instituto de Investigaciones en Bacteriología y Virología Molecular (IBAViM), Facultad de Farmacia y Bioquímica, Universidad de Buenos Aires, Argentina Junín 954, Ciudad de Buenos Aires 1113, Argentina

**Keywords:** rotavirus, G8P[8], Argentina, Bayesian analysis, phylodynamics

## Abstract

Rotavirus is one of the leading causes of diarrhea in children. In 2018, G8P[8], an unusual association of genotypes, was detected with moderate frequency in symptomatic children in Argen-tina, unlike a previous sporadic identification in 2016. The aim of this study was to analyze the dissemination pattern of the G8P[8]-lineage IV strains detected in Argentina. Nucleotide sequences of the VP7 gene of Argentine G8P[8] strains (2016, 2018 and 2019) were studied by discrete phylodynamic analyses, together with other worldwide relevant G8-lineage IV strains. Bayes Factor (BF) was used to assess the strength of the epidemiological association between countries. Phylodynamic analyses determined an evolutionary rate of 3.7 × 10^−3^ (HDP95%: 1.4 × 10^−3^–8.2 × 10^−3^) substitutions/site/year. Likewise, the most recent common ancestor was 32.2 years old, dating back to 1986 (HDP95% = 1984–1988). The spatiotemporal dynamics analysis revealed South Korea as being the country of origin of the Argentine strains (posterior probability of the ancestral state: 0.8471), which was also evidenced by a significant rate of diffusion from South Korea to Argentina (BF: 55.1). The detection of G8 in South America in 2016–2017 was not related to the cases detected in 2018–2019, revealing a new G8 introduction to the region and supporting a transpacific dissemination.

## 1. Introduction

Group A rotavirus represents the main cause of diarrhea in children, with a pronounced impact on morbidity worldwide and mortality in developing countries [1]. Due to its high burden of disease, efforts have been made to develop effective and safe vaccines. In 2006, two oral rotavirus vaccines (Rotarix^TM^ and RotaTeq^TM^) were approved and licensed and demonstrated a significant reduction in specific hospital admissions and deaths [2]. Additionally, two other vaccines were developed and licensed in China (Lanzhou Lamb Rotavirus Vaccine, LLRV) and Vietnam (POLYVAC), but they are only available on the private market in their countries of origin. More recently, two additional vaccines (ROTASIIL^TM^ and ROTAVAC^TM^) have also been prequalified by the World Health Organization (WHO), but they are still mostly used in India [3].

The rotavirus genome comprises 11 double-stranded RNA segments surrounded by a triple protein capsid, and its major evolutionary mechanisms include intergenic reassortment and point mutation, with an average evolutionary rate in the order of 10^−3^ substitutions/site/year [4,5].

Conventionally, rotaviruses are classified in a binary system according to the two outermost capsid protein genes VP7 and VP4, which determine the G-types and P-types, respectively. These proteins give specificity to rotavirus strains and are responsible for eliciting neutralizing antibodies [6]. Although around 41 G-types and 57 P-types have been described to date, only a few that infect humans are considered as usual or common worldwide. Although, theoretically, multiple associations among the G- and P- types in humans are possible, only six of them are the most frequently detected: G1P[8], G2P[4], G3P[8], G4P[8], G9P[8] and G12P[8] [7].

One of the G types considered unusual, the G8 genotype, was endemic in Africa during the last decades of the 20th century and is mostly associated with P[4], P[8] and P[6] genotypes. Recently, G8-lineage IV strains have been detected more regularly in Europe and Asia, associated with P[8] [8,9,10]. In Argentina, G8-lineage V circulated sporadically in 2010, associated with the P[6] genotype in humans, as did lineage II, associated with P[1] and P[14] in animals. However, in 2018, the unusual association G8P[8] was detected with moderate frequency in symptomatic children in our country during post-vaccination surveillance, after a previous sporadic identification in 2016, linked to strains from Chile, all of them belonging to lineage IV [11,12].

While rotavirus universal vaccination has progressed worldwide, a matter of concern is its impact on the emergence of escape mutants or even on the more efficient spread of formerly known unusual strains due to selective pressure. In this scenario, the objective of this study was to analyze the putative origins and dissemination pattern of these G8P[8] strains detected in Argentina.

## 2. Materials & Methods

Phylodynamic analyses were conducted using nucleotide sequences of the VP7 gene of all the G8P[8] strains identified in Argentina and all the G8-lineage IV sequences that were available in GenBank, the Rotavirus Virus Variation Resource from NCBI (https://www.ncbi.nlm.nih.gov/genomes/VirusVariation/Database/nph-select.cgi?taxid=28875, accesed on 20 November 2021), and ViPR (https://www.viprbrc.org/brc/vipr_genome_search.spg?method=ShowCleanSearch&decorator=reo, accesed on 20 November 2021) web sites (n = 52).

The Argentinean G8P[8] dataset comprised five nucleotide sequences from viruses circulating in 2018 that were already reported in our previous study [1], and five other additional VP7 gene sequences from strains that were identified in Argentina and reported in this work. These additional strains (one from 2016, one from 2018 and three from 2019) were detected during a routine molecular surveillance on the rotavirus positive stool specimens of symptomatic children under 5 years of age. Stool samples were submitted to the hospital laboratories of the national network for conventional binary genotyping [11]. VP7 gene was amplified and further sequenced using the Beg9/End9 pair of primers [13].

The evolutionary rate, the time to the most recent common ancestor (tMRCA), and spatial dynamics were determined through the Bayesian Markov Chain Monte Carlo approach implemented in BEAST v1.10.4 [14]. The dataset included a total of 62 sequences from samples obtained from 1988 to 2019. A positive correlation between the genetic divergence and sampling time has been observed using the Root-to-tip analysis with TempEst v1.5.3 [15], suggesting that the dataset is suitable for a phylodynamic analysis with tip dating calibration. The substitution model HKY+I (assessed by the ModelFinder module from the IQ-TREE webserver—http://iqtree.cibiv.univie.ac.at, accesed on 1 December 2021—according to the Bayesian Information Criterion), the Uncorrelated Relaxed Lognormal molecular clock and the GMRF Skyride method for demographic reconstruction were selected as coalescent parameters.

Furthermore, a spatiotemporal process was modeled on time-measured genealogies over discrete sampling locations (countries) using an asymmetric model, and a Bayesian stochastic search variable selection (BSSVS) procedure was applied to obtain the set of spatial diffusion rates that appropriately explained the spatiotemporal process [16]. An analysis was carried out for 150 million generations and evaluated using Tracer software v1.7.1 (http://tree.bio.ed.ac.uk/software/tracer/, accesed on 3 December 2021) to achieve an effective sample size (ESS) of >200, with 10% of the sampling discarded as burn-in. The maximum clade credibility tree (MCCT) was annotated using TreeAnnotator and viewed in FigTree v1.4.4 (http://tree.bio.ed.ac.uk/software/figtree/, accesed on 10 December 2021). Uncertainty in parameter estimates were evaluated in the 95% highest posterior density (HPD95%) interval. In addition, the geographic pattern of dissemination was visualized, and the Bayes Factor (BF) was calculated to weigh the significance of the epidemiological linkage between locations using SpreaD3 v0.9.7rc [17], considering BF > 3 as significant.

The GenBank accession numbers for the five sequences obtained for this study are OM339145-OM339149. The accession numbers of the other Argentinean sequences reported previously by our group [11] and the selected sequences reported worldwide are displayed in Figure 1.

## 3. Results

A total of 62 G8-lineage IV nucleotide sequences from the VP7 gene were included in the dataset, belonging to eleven countries from four regions: Europe (Czech Republic), Africa (Egypt), Asia (China, India, Japan, Singapore, South Korea, Thailand, and Vietnam), and South America (Argentina and Chile).

The evolutionary rate was estimated at 3.7 × 10^−3^ (HDP95% = 1.4 × 10^−3^–8.2 × 10^−3^) nucleotide substitutions/site/year, and the tMRCA was estimated at 32.2 years since 2019, dating back to 1986 (HDP95% = 1984–1988).

Regarding the spatiotemporal dynamics, the analysis revealed that the current G8P[8] strains circulating globally would have originated around 2010 in Thailand (posterior probability of the ancestral state of 0.948). Furthermore, the demographic reconstruction did not reveal any significant change in the effective number of infections of G8-lineage IV over time, possibly showing a still limited diversity and sampling (data not shown).

In South America, the only G8P[8] strain sporadically detected in 2016 in Argentina was associated with the Chilean strains identified at the same period (BF Chile to Argentina = 15.0), as a consequence of a potential introduction from a Southeast Asia country, possibly Japan (Figure 2 and Appendix A). Furthermore, the BF analysis suggested an epidemiological link from Japan to Argentina, with a low supporting value (BF Japan to Argentina = 3.6) (Figure 1 and Figure 2).

On the other hand, South Korea was estimated to be the country of origin of the Argentinean G8P[8] strains detected during the 2018–2019 period, with a posterior probability of the ancestral state of 0.8471 (Figure 1), which was also complemented by a significant epidemiological link from South Korea to Argentina (BF South Korea to Argentina = 55.1) (Figure 2 and Appendix A). Therefore, our analysis showed that the detection of G8-lineage IV cases in South America in 2016–2017 was not related to the cases detected in 2018–2019, since they clustered apart from the 2018/2019 strains, sustained with significant posterior probabilities of the ancestral state, revealing a new G8 introduction to the region and supporting a transpacific dissemination.

## 4. Discussion

G8-lineage IV strains are indicated as being responsible for the recent dissemination worldwide. Even though tMRCA analyses dated the divergence of this lineage back to the 1980s, the scientific reports on its detection were more frequent in the last decade, starting in Southeast Asia. However, the demographic reconstruction of this work was not able to show a recent expansion pattern of this lineage. Additionally, in this study, the evolutionary rate was shown to be similar to what was observed for other G-types considered common, such as G9 and G12 [4,18]. Thus, it seems that some unusual rotavirus genotypes could be currently mimicking the emergence and efficient-spread processes that positioned these two associations as frequent circulating strains. This hypothesis could be explained by considering the progress of universal vaccination that might deplete the population of individuals susceptible to common rotavirus strains and by the fact that heterologous protection might not be complete for certain unusual emerging strains. Thus, further studies are needed to understand what are the key factors that enable some unusual genotypes to gain adaptability advantages over others and cease being endemic in order to circulate globally.

One of the limitations in this study is the limited number of VP7 gene nucleotide sequences of G8-lineage IV strains that were publicly available online, since all the conclusions are based on analyses on the sequences that could be included. Therefore, underrepresentation could be observed in some particular regions that do not have a continuous surveillance system or that have detected G8P[8] strains in the last decade but whose nucleotide sequences are not available online [19,20].

Nevertheless, all things considered, our data strongly suggest that the introduction of the Argentinean G8P[8] strains that circulated at a moderate frequency in 2018–2019 occurred via South Korea as a consequence of a prior regional spread in Southeast Asia. Conversely, the previous strain detected contemporaneously in Chile in 2016 resembled an independent introduction, possibly from Japan. Our conclusions arise from the combination of the MCCT and BF analyses. In this manner, although different hypotheses can be raised based on the results obtained from BFs, those with BF >10 might offer stronger evidence than those with moderate support (BF = 3–5).

We underscore these types of studies because they provide significant evidence about the patterns of emergence, viral evolution, and spread of unusual rotavirus strains worldwide in the post-vaccination era.

## Figures and Tables

**Figure 1 viruses-14-02223-f001:**
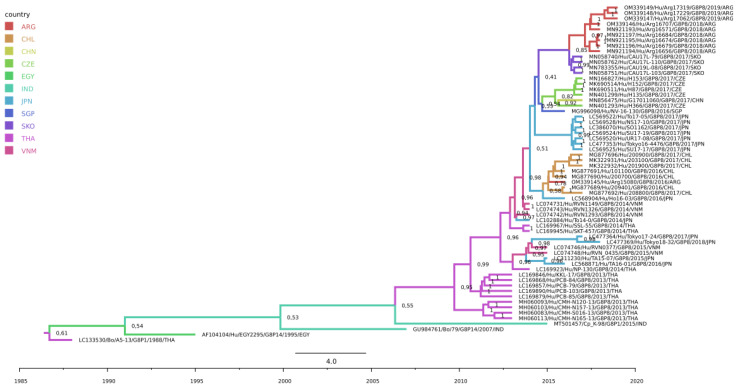
Maximum Clade Credibility Tree (MCCT) of G8-lineage IV strains. Branches are color-coded according to their location’s discrete state (ARG: Argentina, CHL: Chile, CHN: China, CZE: Czech Republic, EGY: Egypt, IND: India, JPN: Japan, SGP: Singapore, SKO: South Korea, THA: Thailand, VNM: Vietnam). Timescale is indicated below the tree. Posterior probability values of the ancestral state are shown in each branch. Additionally, clade posterior probability values for groups with Argentinean sequences are shown at nodes in parentheses.

**Figure 2 viruses-14-02223-f002:**
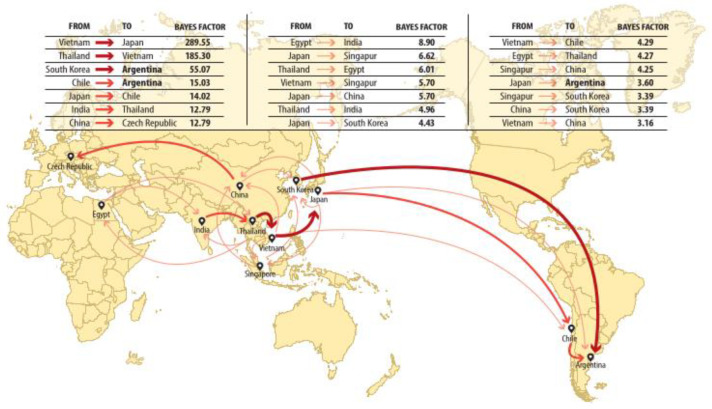
Global migration pattern of G8-lineage IV strains. Linkage points between countries are indicated from/to their center, and the width of the connection is proportional to the Bayes factor estimation according to the reference scale (only BF > 3.0 are displayed).

## Data Availability

The GenBank accession numbers for the five sequences obtained for this study are OM339145-OM339149.

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
