# Peer review of "Phylogeography of Rotavirus G8P[8] Detected in Argentina: Evidence of Transpacific Dissemination"

_viruses, 2022, doi:10.3390/v14102223_

Round 1

Reviewer 1 Report

In the manuscript of Degiuseppe et al. “Phylogeography of rotavirus G8 detected in Argentina: evidence of transpacific dissemination”, the authors describe the pattern of geographic dissemination of Argentinean G8 lineage IV strains. Despite being an interesting study, the major weakness is related to the novelty and the possible impact of this manuscript. It should have been an additional analysis of the previously published article (doi: 10.1016/j.meegid.2021.104940). To deserve publication in Viruses (IF 5.8) it should be strengthened in order to maintain the journal reputation; you may expand the analysis to the worldwide dissemination and not only put the focus in Argentina. Some additional concerns are detailed below.

- The title does not represent the study. The G8 are not all the detected in Argentina, but only the G8 lineage IV associated with P8.

- There is a know concern about the temporal structure of RVA strains. This was not assessed and should be done to be sure that the strains used have it, in order to validate the phylodynamic analysis.

- The authors mention that some strains were obtained in this study. How were obtained? There is no mention of the methodology. How was performed the sampling? Were from symptomatic/asymptomatic patients? There should be an extensive description about this.

- The limited sampling should be better discussed, since probably there are strains that were not detected/sequenced/uploaded to the database, and the results shown may be somewhat different from the reality (i.e. only used sequences from Czech Republic of Europe and any sequences from North America).

- “it was detected more frequently in the last decade due to an expansion pattern”. This is not in consonance with the reported in results section "the demographic reconstruction did not reveal any significant change in the effective number of infections of G8-lineage IV over time".

- In Figure 2, there are three possible introductions to Argentina. One from South Korea, one from Chile and one from Japan. However, there is only one Argentinean sequence (2016) related to Japan and Chile. The authors hypothesize that the introduction is from Chile, with an ancestral origin in Japan, but how the authors explain the two introduction from Chile (BF=15) and Japan (BF=3.6) determined by the analysis and the BF values?

Author Response

Reviewer #1

In the manuscript of Degiuseppe et al. “Phylogeography of rotavirus G8 detected in Argentina: evidence of transpacific dissemination”, the authors describe the pattern of geographic dissemination of Argentinean G8 lineage IV strains. Despite being an interesting study, the major weakness is related to the novelty and the possible impact of this manuscript. It should have been an additional analysis of the previously published article (doi: 10.1016/j.meegid.2021.104940). To deserve publication in Viruses (IF 5.8) it should be strengthened in order to maintain the journal reputation; you may expand the analysis to the worldwide dissemination and not only put the focus in Argentina. Some additional concerns are detailed below.

We appreciate very much and thank both reviewers for the observations and questions that make us improve the presentation and discussion of our results.

Below you will find the response to each comment. 

- The title does not represent the study. The G8 are not all the detected in Argentina, but only the G8 lineage IV associated with P8.

The title was modified accordingly with the study.

- There is a know concern about the temporal structure of RVA strains. This was not assessed and should be done to be sure that the strains used have it, in order to validate the phylodynamic analysis.

Unfortunately, we omitted to describe the evaluation of the temporal signal in the first version of the manuscript. However, this analysis was performed with the TempEst software prior to the phylodynamic analysis. Below is a screenshot of that analysis, showing a positive trend in the root-to-tip regression analysis.

This information was included in the revised version of the manuscript.

- The authors mention that some strains were obtained in this study. How were obtained? There is no mention of the methodology. How was performed the sampling? Were from symptomatic/asymptomatic patients? There should be an extensive description about this.

In the first version of the manuscript, we described the origin of the new strains introduced in this work in the Introduction section to stress that this work was focused on phylodynamic analyses. However, considering the reviewer´s observation, some clarification was included in the Materials and Methods section of the revised version of the manuscript.

Briefly, five new strains were sequenced (one from 2016, one from 2018 and three from 2019), after being detected during routine molecular surveillance on the rotavirus-positive stool specimens from symptomatic children under 5 years of age that hospital laboratories from the national network submitted for conventional binary genotyping. Thus, the manuscript was modified to clarify this point.

- The limited sampling should be better discussed, since probably there are strains that were not detected/sequenced/uploaded to the database, and the results shown may be somewhat different from the reality (i.e. only used sequences from Czech Republic of Europe and any sequences from North America).

As the reviewer pointed out, the phylodynamic results and conclusion reached depend on the dataset, and we certainly share the reviewer´s concern about the possible limited representation of the virus diversity in the database.

This study included all G8-lineage IV sequences that were available in GenBank, Rotavirus Variation Resource and ViPR, and we are aware that the limited number of available sequences is one of the main limitations of our study. However, in spite of these limitations, we believe that this should not prevent trying to deepen the study of the evolutionary history of the lineage, but serve as an initial point of discussion up to more sequences are available and new analyses can be performed.

Finally, we have stressed this limitation in the Discussion section of the manuscript.

- “it was detected more frequently in the last decade due to an expansion pattern”. This is not in consonance with the reported in results section "the demographic reconstruction did not reveal any significant change in the effective number of infections of G8-lineage IV over time".

The phrase mentioned by the reviewer comes from speculation from reports about G8-lineage IV circulation in recent times. However, we agree with the reviewer that our demographic reconstruction analysis was not able to show an expansion pattern of the lineage. Therefore, the text was modified accordingly.

- In Figure 2, there are three possible introductions to Argentina. One from South Korea, one from Chile and one from Japan. However, there is only one Argentinean sequence (2016) related to Japan and Chile. The authors hypothesize that the introduction is from Chile, with an ancestral origin in Japan, but how the authors explain the two introduction from Chile (BF=15) and Japan (BF=3.6) determined by the analysis and the BF values?

Thank you for your comment. Our conclusions arise from the combination of the MCCT and the BFs analyses. In 2016 we identified only one G8P[8] strain during our regular surveillance activities. On the other hand, the MCCT hypothesis clusters this 2016 strain with the Chilean G8-lineage IV strains, but apart from the Argentinean 2018/2019 which have a different putative origin, supported with significant posterior probability of the ancestral state values. Also, although different hypotheses can be raised based on the results obtained from BFs, those with BF >10 might offer stronger evidence than those with moderate support (BF=3-5).

Despite being expected that the state transitions in the MCCT would be also represented by the corresponding well-supported rates, sometimes this is not the case, especially for states with a low number of sequences in particular clades.

The origin of the discrepancy mentioned by the reviewer may be tracked in the fact that Bayes factors to identify well-supported rates between locations rise from the posterior distribution (of trees and parameters) in a convergent MCMC, and therefore, the information about a set of parameter and topologies -and not from individual results- is kept. However, the MCCT (Figure 1) is only one tree of the posterior distribution of trees, selected for showing the summed information of the overall distribution of results. Values of the probabilities of the ancestral states shown in the MCCT (Figure 1) may indicate that other ancestral locations than that shown as the most probable could locate the internal configuration of the group Arg2016/CHL/JPN, and maybe sometimes supporting transition JPN to Arg2016 more directly (instead of CHL to ARG).

Nevertheless, we addressed this point in the revised version of the manuscript for clarification purposes and included the MCCT with clade posterior probability values as supplementary file.

Reviewer 2 Report

The work describes the phylodynamic of an emergent G8P[8] rotavirus in Argentina. The study detects the introduction of this strain in two opportunities from Chile and the latter from Korea. This type of study is essential for epidemiology and vaccine coverage and update.

Minor corrections 

In the introduction if you mention the vaccines in India, maybe the vaccine applied in China is worthy to be mentioned.

In the listed RVA strain circulating in humans why you did not included the variant carrying P[6] genotypes that are also important?

Results

There are no G8P[8] RVA strains available from Brasil and Uruguay to add to the analysis? 

Figure 2, maybe will be more clear to add the dates of the transmission to make clear that two transmission event were studied 

Author Response

Reviewer #2

The work describes the phylodynamic of an emergent G8P[8] rotavirus in Argentina. The study detects the introduction of this strain in two opportunities from Chile and the latter from Korea. This type of study is essential for epidemiology and vaccine coverage and update.

We appreciate very much and thank both reviewers for the observations and questions that make us improve the presentation and discussion of our results.

Below you will find the response to each comment.

Minor corrections

In the introduction if you mention the vaccines in India, maybe the vaccine applied in China is worthy to be mentioned.

The manuscript was modified accordingly.

In the listed RVA strain circulating in humans why you did not included the variant carrying P[6] genotypes that are also important?

In this study we did not find any G8-lineage IV associated with P[6] genotype reported worldwide. As mentioned in the Introduction section, in Argentina only two sporadic samples bearing G8P[6] association were detected in 2010. However, these G8 strains belonged to lineage V.

Results

There are no G8P[8] RVA strains available from Brasil and Uruguay to add to the analysis?

Unfortunately, although there are very limited reports of sporadic detection of G8P[8] strains in Brazil, there were not any sequences available for phylodynamic analysis. This is pointed out in the Discussion section as one of the limitations of these types of studies.

Figure 2, maybe will be more clear to add the dates of the transmission to make clear that two transmission event were studied

The phylogeographic analysis integrates the information of the whole dataset in all the periods included in the sequences analyzed. As we pointed out in response to reviewer #1, those results arise from considering the posterior distribution of the hypothesis (trees and parameters) visited by the MCMC. So, every transition supported by BF analysis actually may represent one or many linkages between locations. We clarify this point in the revised version of the manuscript.

Round 2

Reviewer 1 Report

The authors have addressed the concerns, the manuscript can be accepted.

As a minor comment, I could not find the clade posterior probability values for groups with Argentinean sequences shown at nodes in parenthesis (lines 124-125).